# Mapping QTLs for enhancing early biomass derived from *Aegilops tauschii* in synthetic hexaploid wheat

Yumin Yang[1,2,3ʘ], Hongshen Wan[3,4ʘ], Fan Yang[1], Chun Xiao[2], Jun Li[3,4], Meijin Ye[1], Chunxiu Chen[2], Guangmin Deng[2], Qin Wang[3,4], Aili Li[5], Long Mao[5], Wuyun Yang[ID][3,4]*, Yonghong Zhou[1]*

1 Triticeae Research Institute, Sichuan Agricultural University, Chengdu, China, 2 Soil and Fertilizer Research Institute, Sichuan Academy of Agricultural Sciences, Chengdu, China, 3 Key Laboratory of Wheat Biology and Genetic Improvement on Southwestern China (Ministry of Agriculture and Rural Areas), Chengdu, China, 4 Crop Research Institute, Sichuan Academy of Agricultural Sciences, Chengdu, China, 5 Institute of Crop Sciences, Chinese Academy of Agricultural Sciences, Beijing, China

ʘ These authors contributed equally to this work.
* yangwuyun@126.com (WY); zhouyhdavid@126.com (YZ)

**Data Availability Statement:** All relevant data are within the manuscript and its Supporting Information files.

## Abstract

Strong early vigour plays a crucial role in wheat yield improvement by enhancing resource utilization efficiency. Synthetic hexaploid wheat (SHW) combines the elite genes of tetraploid wheat with *Aegilops tauschii* and has been widely used in wheat genetic improvement for its abundant genetic variation. The two SHWs Syn79 and Syn80 were derived from the crossing of the same tetraploid wheat DOY1 with two different *Ae. tauschii* accessions, AT333 and AT428, respectively. The Syn80 possessed better early vigour traits than Syn79, theretically caused by their D genome from *Ae. tauschii*. To dissect their genetic basis in a hexaploid background, 203 recombinant inbred lines (RILs) derived from the cross of Syn79 x Syn80 were developed to detect quantitative trait loci (QTL) for four early biomass related traits: plant height (PH), tiller number (TN), shoot fresh weight (SFW) and shoot dry weight (SDW) per plant, under five different environmental conditions. Determined from the data of SNP markers, two genome regions on 1DS and 7D were stably associated with the four early biomass related traits showing pleiotropic effects. Four stable QTLs *QPh. saas-1DS*, *QTn.saas-1DS*, *QSfw.saas-1DS* and *QSdw.saas-1DS* explaining 7.92, 15.34, 9.64 and 10.15% of the phenotypic variation, respectively, were clustered in the region of 1DS from *AX-94812958* to *AX-110910133*. Meanwhile, *QPh.saas-7D*, *QTn.saas-7D*, *QSfw. saas-7D* and *QSdw.saas-7D* were flanked by *AX-109917900* and *AX-110605376* on 7D, explaining 16.12, 24.35, 15.25 and 13.37% of the phenotypic variation on average, respectively. Moreover, these genomic QTLs on 1DS and 7D enhancing biomass in the parent Syn80 were from *Ae. tauschii* AT428. These findings suggest that these two QTLs from *Ae. tauschii* can be expressed stably in a hexaploid background at the jointing stage and be used for wheat improvement.

**Funding:** This work was financially supported by National Natural Science Foundation of China (Grant No. 31661143007, 31401383) and Department of Science and Technology of Sichuan Province (Grant No. 2017JY0077, 2017JY0286, 2020YJ0469).

**Competing interests:** The authors have declared that no competing interests exist.

## Introduction

Common wheat (*Triticum aestivum* L., 2n = 6x = 42, AABBDD), which is an important food crop throughout the world, originated from the spontaneous hybridization of tetraploid *Triticum turgidum* wheat (2n = 4x = 28, AABB) with diploid *Aegilops tauschii* Coss (2n = 2x = 14, DD) [1,2]. It is believed that only a few accessions of the donor species were involved in the evolution of common wheat, especially for the D genome donor. Consequently, the genetic diversity of common wheat decreased significantly compared to its donor species. Due to this evolutionary bottleneck, most of the genetic variation in *Ae. tauschii* did not exist in the commonly available hexaploid germplasm [3], and only 7% of the variants observed in *Ae. tauschii* were reserved in common wheat [4,5]. To enhance the transferal efficiency of elite genes from *Ae. tauschii* species to common wheat, scientists created synthetic hexaploid wheat (SHW) from crosses between *T. turgidum* and *Ae. tauschii* to broaden the genetic variation of hexaploid wheat [6]. Over 1000 SHW lines were produced by using more than 600 *Ae. tauschii* accessions stored at the International Maize and Wheat Improvement Center (CIMMYT; Mexico City, Mexico) [7]. SHWs with their vast genetic diversity have shown outstanding superiority in resistance to diseases and pests, tolerance to environmental stresses, and desirable quantitative traits, so these have been used widely in common wheat breeding [8–14]. Chinese scientists have shown a high interest in CIMMYT SHW lines since the early 1990s [15–18]. More than 200 CIMMYT SHW accessions were introduced into China in 1995 [14]. In recent years, several commercial wheat varieties have also been created and released in China [9,14,19]. In addition, several favourable introgressions from *Ae. tauschii* have been identified in synthetic derivatives [19]. A major QTL on 4DL associated with leaf sheath hairiness in a synthetic derivative of the wheat variety Chuanmai42 was identified, and its wild allele was found to have originated from *Ae. tauschii*, which has significantly increased grain weight, grain yield, and yield-related characters [20].

Vigorous cultivars have advantages for enhancing the population's water-use efficiency by providing shade to the soil surface faster and thereby reducing evaporative losses from the soil [21–23]. Rapid early development of leaf area and the root system are associated with increased water and nutrient use efficiency, high rates of light interception and biomass production resulting in drought tolerance and high yield potential [22,23]. In recent years, we have screened CIMMYT SHWs for high biomass and found two SHWs (Syn79 and Syn80) derived from the same tetraploid wheat (durum wheat DOY1), with two different *Ae. tauschii* accessions, which have significantly different biomass during the entirety of the development stage. We attributed the significant difference in biomass between the two SHWs to the different genotypes in the two D genome donors. The vegetative growth, nutrient accumulation, nutrient distribution and utilization of Syn79 and Syn80 were significantly different under different environmental conditions [24]. To evaluate the genetic impact of the different D genomes on early vigour in hexaploid wheat, a population of recombinant inbred lines (RILs) derived from a cross between Syn79 and Syn80 was developed. The goal of this study was to map the major QTLs associated with early biomass accumulation contributed from *Ae. tauschii* in a hexaploid wheat background at jointing stage for the molecular breeding of wheat yield using SHWs.

## Materials and methods https://dx.doi.org/10.17504/protocols.io. bgrnjv5e

### Plant materials

Two hundred and three $F_9$ recombinant inbred lines (RILs) derived from a Syn79 x Syn80 cross and their parents were used for QTL mapping in this study. Syn79 and Syn80 were

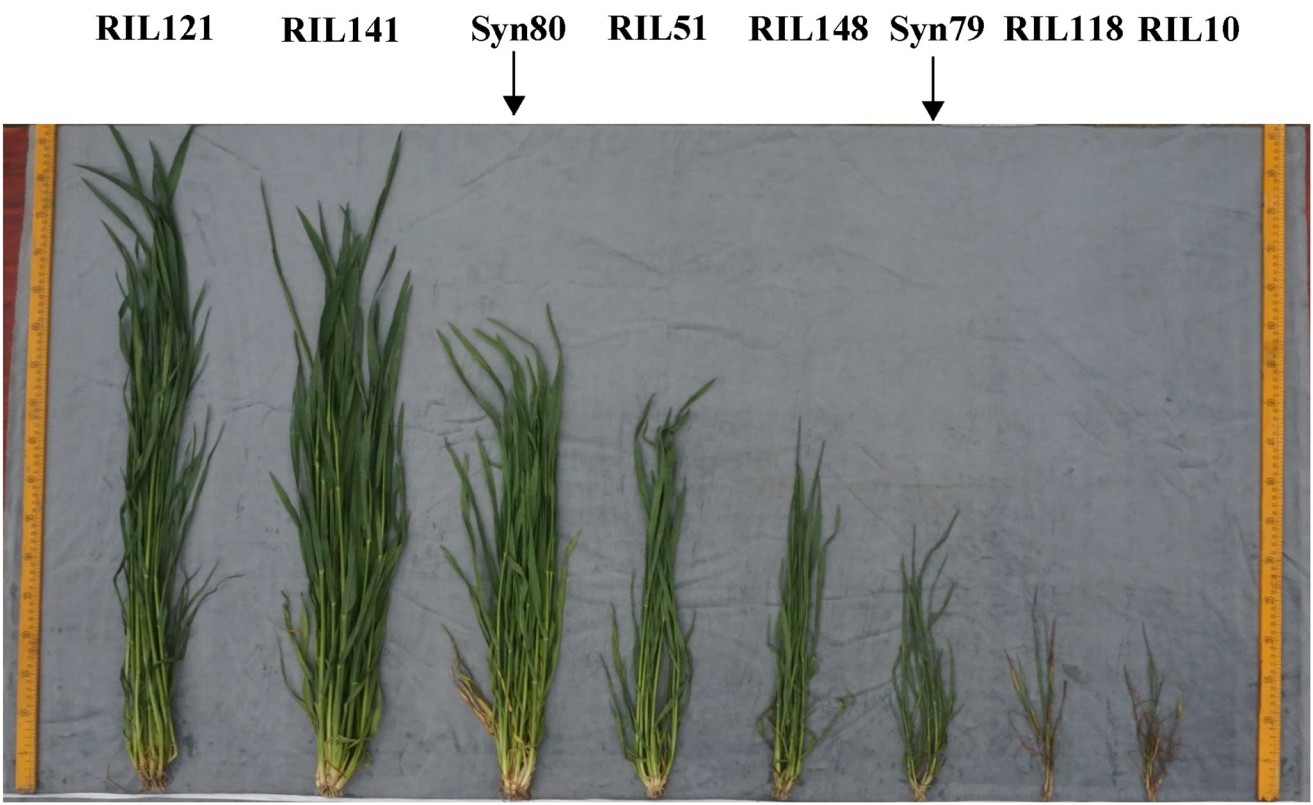

**Fig 1. Early growth of the two parents and their RILs in the jointing stage.**

generated from durum wheat DOY1 (2n = 28, AABB) crossed with *Ae. tauschii* (2n = 14, DD) by CIMMYT [6]. A and B genomes of Syn79 and Syn80 were from the same durum donor DOY1, while their D genomes were from two different *Ae. tauschii* accessions (AT333 and AT428). Syn80 had stronger early vigour than Syn79 (Fig 1), due to their different D genomes, and AT428 possessed better early vigour traits than AT333.

## Field trials

A total of five trials for Syn79, Syn80 and 203 RILs were conducted at Guang-Han Station (GHS) in 2017–2019 (2017GHS, 2018GHS, 2019GHS) and Cang-Shan Station (CSS) in 2017 and 2018 (2017CSS, 2018CSS). Both stations are members of the Sichuan Academy of Agricultural Sciences (SAAS). GHS and CSS are representative of the plains and hilly regions in Sichuan province, respectively. The chemical properties of the soil at these sites from five trials are shown in Table 1. The organic matter, total nitrogen and available nitrogen of the soil in GHS were all significantly higher than that in CCS, and the total potassium of the soil in CSS was more than that in GHS (Table 1).

## Trait evaluation

Four early biomass related traits, plant height (PH), tiller number (TN), shoot fresh weight (SFW) and shoot dry weight (SDW) per plant were investigated in the RILs and their parents at the jointing stage. The phenology and growing periods of the two parents and the RILs were only slight different, their phenotypic data were collected at one time when the first internode

**Table 1. Chemical properties of soil in different field trials.**

| Trials | pH | Organic matter (g/kg) | Total nitrogen (g/kg) | Total phosphorus (g/kg) | Total potassium (g/kg) | Available nitrogen (mg/kg) | Available phosphorus (mg/kg) | Available potassium (mg/kg) |
|---|---|---|---|---|---|---|---|---|
| 2017GHS | 6.84 | 31.9 | 1.99 | 0.723 | 16.25 | 165 | 6.9 | 90 |
| 2018GHS | 6.45 | 39.7 | 2.31 | 0.860 | 18.77 | 206 | 15.8 | 105 |
| 2019GHS | 6.71 | 28.9 | 2.03 | 0.674 | 19.00 | 183 | 11.0 | 96 |
| 2017CSS | 7.81 | 9.5 | 0.77 | 0.556 | 23.81 | 47 | 3.7 | 100 |
| 2018CSS | 8.24 | 15.7 | 1.23 | 0.328 | 22.90 | 97 | 2.9 | 137 |

The trials were performed in randomized complete blocks with three replicates. Each plot had five 1.5 m rows spaced 0.5 m apart. At the two-leaf stage, only ten evenly distributed plants in each row were retained for further growth. Field management consisted of commonly under-taken practices in wheat production.

came out about 110 days after sowing. In each plot, 10 plants were randomly selected to evaluate traits associated with early biomass, dislodging plants at the ends of each row avoiding within-row edge effects. PH and TN were investigated in the field, which was finished within 1–2 days under the same trial environment. Then the shoots of these 10 sampled plants were cut for measuring SFW and SDW. SFW was accomplished within 12 hours after sampling. When measuring SDW, the separated shoot was dried to a constant weight at 65 ˚C after 10-minute exposure to 120 ˚C. All traits were described based on the mean values of 10 plants in each corresponding row.

## SNP genotyping

A total of 50 mg of fresh plant leaves was collected from 2-week-old seedlings and DNA was extracted using the NuClean Plant Genomic DNA Kit (CWBio, Beijing, China). Eluted DNA was quantified using a Qubit 4 Fluorometer (Life Technologies Holdings Pte Ltd, Singapore) and then normalized using a 12-channel electronic pipette with a volume ranging from 10 to 100μL (Eppendorf, Hamburg, Germany) to obtain the concentration required for genotyping.

The RILs and their parents, Syn79 and Syn80, were genotyped on the Affymetrix platform of the Axiom Wheat Breeder's Genotyping Array with 13947 SNP markers including 1272 functional markers by China Golden Marker Biotech Co Ltd (Beijing, China). The collected fluorescence signal from the SNP array processed and analyzed using functions in the apt-genotype-axiom for genotype calling, ps-metrics for generating various QC metrics and ps-classification for classifying SNPs in the software of Affymetrix Axiom Analysis Suite version 4.0.1. Among 13947 SNP markers, a total of 3480 SNPs were distributed on the D genome and were used for parental polymorphism analysis.

## Statistical and QTL analysis

Descriptive analyses, analysis of variance (ANOVA) and correlation analyses for the phenotypic data were calculated using the SPSS statistical package (SPSS Inc., Chicago, IL). Variation of genotypes for phenotypic traits was evaluated using mean, standard deviation (SD), the coefficient of variation (CV), maximum (Max) and minimum (Min). An ANOVA was calculated for all traits based on a general linear model (GLM) to detect the effect of genotypes, environments and genotype × environment interactions. Broad sense heritability ($H^2$) was estimated with the formula: $H^2 = \sigma^2 g / (\sigma^2 g + \sigma^2 ge/n + \sigma^2 e/nr)$, where $\sigma^2 g$ is the genetic variance, $\sigma^2 ge$ is the variance of the genotype-environment interaction, $\sigma^2 e$ is the experimental error variance, n is the number of trials and r is the number of replications.

The QTL IciMapping Software version 4.1 [25,26] was used for genetic linkage map construction. The location of the SNP marker was aligned according to the physical map of

*Ae. tauschii* AL8/78 for the D genome [27]. The genetic linkage map was constructed according to 153 polymorphic markers between Syn79 and Syn80 (the parents), which were screened from 3480 SNP markers distributed on the D genome. The map covered over 803.84 cM on the wheat D genome, with an average distance of 5.25 cM between adjacent polymorphic markers.

QTL analyses for the measured traits under the five different environmental conditions were performed using the inclusive composite interval mapping (ICIM) option on the QTL IciMapping Software version 4.1. The significant LOD threshold was determined by 1000 permutations and a significance threshold of P = 0.05. Linked QTLs with genetic distances of less than 20 cM were considered as one single QTL, which were named according to Ayalew et al. [28].

## Results

### Phenotypic analysis

Five different field trials were conducted at two locations over 3 years to evaluate early biomass related traits of the RIL population as well as their parents Syn79 and Syn80. Syn80 had greater early biomass than Syn79 (Fig 1). The values of PH, TN, SFW and SDW for Syn80 were significantly larger than those of Syn79 under all five environmental conditions (Table 2). Independent of the differences between the two parents, in all trials there was significant variation in the investigated traits of the RIL populations, with values spanning much larger ranges than

**Table 2. Parental values, population distribution parameters, and heritability of the investigated traits.**

| Trait | Environment | Parents | | RILs | | | H$^2$ | F-values from ANOVA | | |
|---|---|---|---|---|---|---|---|---|---|---|
| | | Syn79 | Syn80 | Mean±SD | CV(%) | Min-Max | (%) | Environment | Genotype | Environment×genotype |
| PH | 2017GHS | 31.87 | 48.64** | 38.43±5.98 | 15.56 | 27.00–59.20 | 43.27 | 1368.74** | 14.05** | 2.02** |
| (cm) | 2018GHS | 34.48 | 53.81** | 50.12±8.07 | 16.10 | 29.38–65.33 | | | | |
| | 2019GHS | 46.11 | 63.33** | 60.87±8.20 | 13.47 | 38.11–81.94 | | | | |
| | 2017CSS | 42.00 | 56.33** | 54.40±8.34 | 15.33 | 27.17–69.33 | | | | |
| | 2018CSS | 49.67 | 65.11** | 59.84±9.07 | 15.16 | 27.58–76.56 | | | | |
| TN | 2017GHS | 5.60 | 12.53** | 10.01±2.58 | 25.77 | 4.50–17.90 | 43.11 | 780.24** | 14.32** | 2.05** |
| (No./plant) | 2018GHS | 8.17 | 16.67** | 15.05±3.92 | 26.05 | 6.00–23.00 | | | | |
| | 2019GHS | 10.00 | 15.78** | 14.23±3.73 | 26.21 | 5.78–21.67 | | | | |
| | 2017CSS | 7.00 | 12.00** | 9.22±2.58 | 27.98 | 2.00–15.00 | | | | |
| | 2018CSS | 8.33 | 13.22** | 11.16±3.31 | 29.66 | 2.33–18.33 | | | | |
| SFW | 2017GHS | 17.09 | 73.72** | 48.25±25.60 | 53.06 | 7.02–136.54 | 40.20 | 144.40** | 14.05** | 2.16** |
| (g/plant) | 2018GHS | 21.79 | 70.78** | 60.92±29.91 | 49.10 | 7.30–156.94 | | | | |
| | 2019GHS | 29.40 | 78.37** | 64.94±29.89 | 46.03 | 10.97–169.70 | | | | |
| | 2017CSS | 26.58 | 82.57** | 52.69±25.50 | 48.40 | 3.59–132.55 | | | | |
| | 2018CSS | 36.89 | 96.08** | 75.44±39.27 | 52.05 | 4.72–158.86 | | | | |
| SDW | 2017GHS | 2.85 | 10.22** | 6.64±3.34 | 50.30 | 1.04–16.19 | 39.20 | 161.47** | 84.08** | 6.60** |
| (g/plant) | 2018GHS | 3.43 | 10.24** | 8.35±3.58 | 42.87 | 1.04–20.54 | | | | |
| | 2019GHS | 4.63 | 11.87** | 8.96±4.31 | 48.10 | 1.43–17.80 | | | | |
| | 2017CSS | 4.43 | 11.45** | 7.25±3.28 | 45.24 | 0.46–18.13 | | | | |
| | 2018CSS | 5.59 | 13.98** | 10.18±4.95 | 48.62 | 0.69–20.57 | | | | |

* And ** indicate significant differences at P = 0.05 and 0.01, respectively. PH: plant height, TN: tiller number, SFW: shoot fresh weight, SDW: shoot dry weight, SD: standard deviation, CV: the coefficient of variation, Max: maximum, Min: minimum, RILs: recombinant inbred lines, H$^2$: broad sense heritability, ANOVA: analysis of variance.

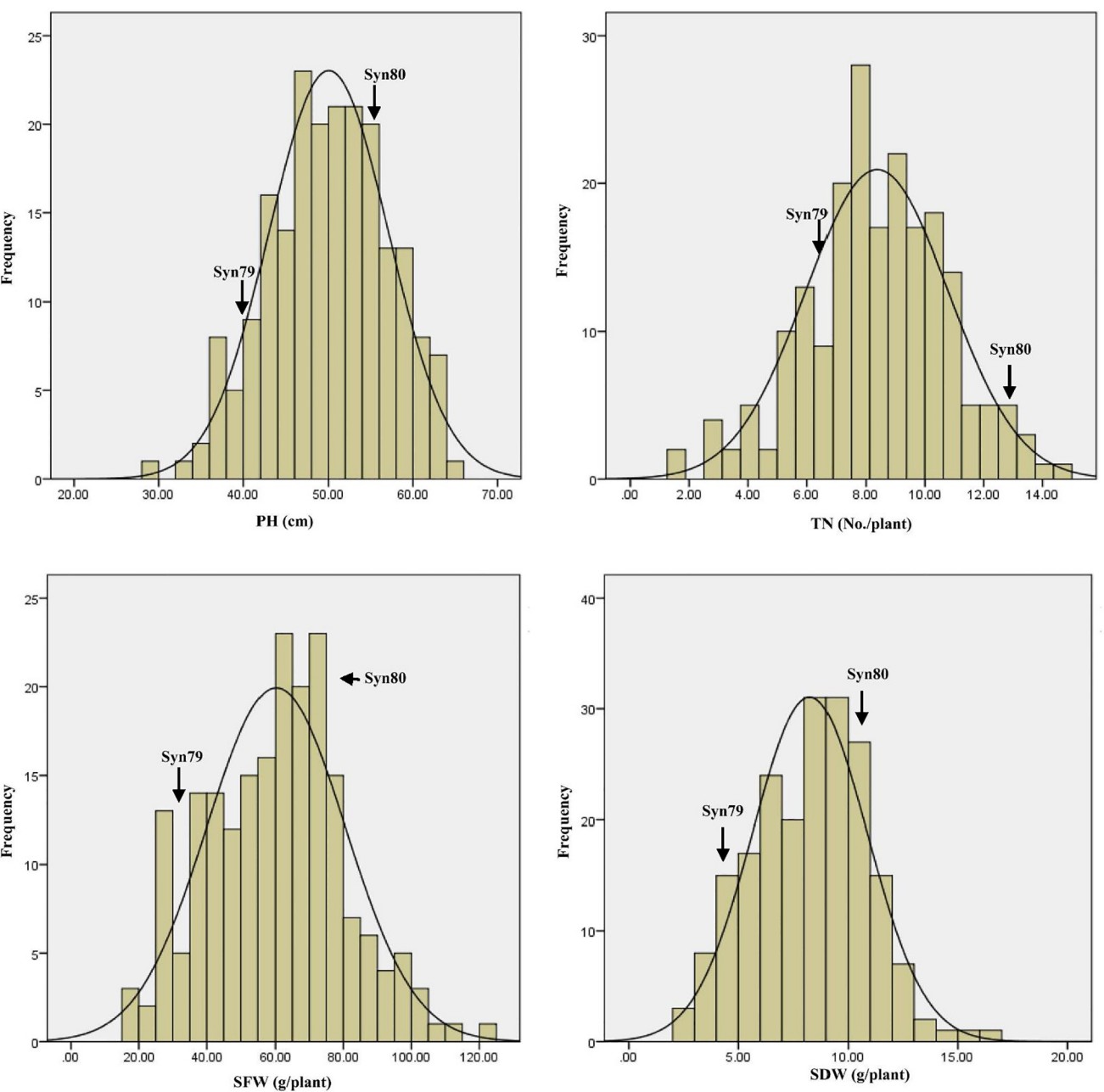

**Fig 2. Distribution graph of the phenotypic data for plant height (PH), tiller number per plant (TN), shoot fresh weight (SFW) and shoot dry weight (SDW) under five different environments.**

those defined by the parental values. The phenotypic data were normally distributed in the RILs (Fig 2). Variation in the phenotypic data was tremendous in the RILs, especially for SFW and SDW. Variation was determined by genotype, environment and genotype × environment interactions. Their heritabilities ranged from 39.20 to 43.27% (Table 2). This suggested that those phenotypic traits were controlled by multiple genes and also significantly affected by the environment.

Correlations among PH, TN, SFW and SDW in each trial are presented in Table 3. This shows that significant positive correlations among these traits were detected in the early

**Table 3. Correlation coefficients between the four traits in RILs in different trials.**

| | TN | SFW | SDW |
|---|---|---|---|
| PH | 0.673** | 0.724** | 0.733** |
| | 0.424** | 0.364** | 0.355** |
| | 0.683** | 0.784** | 0.808** |
| | 0.606** | 0.821** | 0.835** |
| | 0.612** | 0.791** | 0.791** |
| TN | | 0.708** | 0.711** |
| | | 0.142* | 0.141* |
| | | 0.731** | 0.728** |
| | | 0.668** | 0.678** |
| | | 0.749** | 0.713** |
| SFW | | | 0.967** |
| | | | 0.969** |
| | | | 0.978** |
| | | | 0.983** |
| | | | 0.980** |

*And** indicate significance at P = 0.05 and 0.01 level, respectively. PH: plant height, TN: tiller number, SFW: shoot fresh weight, SDW: shoot dry weight.

growth stage. The average coefficients in the five trials ranged from 0.581 to 0.975. PH was significantly positively correlated with TN, and the coefficients ranged from 0.424 to 0.683 across each trial (Table 3). Both PH and TN showed significant positive correlations with SFW and SDW, and the coefficients were higher than that between PH and TN. These results suggest that greater early biomass is related with higher PH, more TN, heavier SFW and SDW.

### Genetic map of the D genome

In this study, we used a Wheat Breeder's Genotyping Array to genotype the A, B and D genomes. For the A and B genomes, a total of 10467 SNP labels anchored on the genotyping array were used to check the genotype of the A and B genomes in the RIL population and their parents, which were generated from the same A and B genomes'donor. The results showed that almost all SNP markers on the A and B genomes had no polymorphism between the two parents. For the D genome, 3480 SNP labels were selected to fix on the chip by China Golden Marker Biotech Co Ltd (Beijing, China). Among these scanned markers, 153 markers on the D genome had polymorphism between the two parents, which were unequally distributed on the seven chromosomes of the D genome (Table 4). The number of polymorphic markers on different chromosomes ranged from 8 on 3D to 34 on 7D (Table 4).

For linkage map construction, SNP markers were grouped according to their anchored chromosomes in the *Ae. tauschii* AL8/78 D genome, and then aligned by the nnTwoOpt method [25–27]. The entire genetic map covered over 803.84 cM of the D genome with an average distance between adjacent markers of 5.25 cM (Table 4). The average distance between two adjacent markers ranged from 2.67 cM to 6.93 cM. For all of the 7 chromosomes, the linkage maps ranged from 21.35 cM to 214.50 cM. For the chromosomes 2D and 3D, the total distances of the constructed linkage maps in this population and the Wheat Breeder's Genotyping Array were 69.22 cM and 21.35 cM, respectively. Out of the genomic regions of the linkage maps, no polymorphic markers were detected by this SNP array.

**Table 4. SNP markers on the D genome.**

| Parameter | 1D | 2D | 3D | 4D | 5D | 6D | 7D | Total |
|---|---|---|---|---|---|---|---|---|
| Total Makers | 370 | 634 | 536 | 265 | 550 | 428 | 697 | 3480 |
| Polymorphic markers | 30 | 20 | 8 | 21 | 25 | 15 | 34 | 153 |
| Polymorphism rate (%) | 8.11 | 3.15 | 1.49 | 7.92 | 4.55 | 3.50 | 4.88 | 4.40 |
| Map length (cM) | 127.58 | 69.22 | 21.35 | 145.52 | 135.72 | 89.95 | 214.50 | 803.84 |
| Distance between polymorphic markers (cM) | 4.25 | 3.46 | 2.67 | 6.93 | 5.43 | 6.00 | 6.31 | 5.25 |

Genotypic markers were tested for segregation distortion (deviation from the expected 1:1 ratio) by Chi-squared tests. Among the 153 SNP loci, 54 loci showed segregation distortion in RILs (Table 5). Almost all loci were biased to Syn80, showing larger early biomass, which means that in those loci most of the progeny RILs preferentially inherited the female parent Syn80. Only four loci were male-biased (Table 5). Among those female-biased loci, the number of loci on the different chromosomes were distributed from 1 on 3D to 18 on 7D. Three genomic regions were detected as Syn80-biased on chromosome 1D, 2D and 7D (Table 5), and these covered about 50, 20 and 40 cM on 1D, 2D and 7D, respectively.

## QTLs on the D genome

With the linkage map constructed by 153 SNP markers on the D genome, QTLs for PH, TN, SFW and SDW were identified under five environmental conditions using the inclusive composite interval mapping program (ICIM).

PH and TN are common agronomic traits in wheat, and the higher PH and TN in the seedling growth stage were positively correlated with the enhanced water-use efficiency of the population due to the soil surface being shading faster, which reduces evaporative losses from the soil. A total of two QTLs for PH were identified on chromosome 1DS and 7D (Table 6; Fig 3). The QTL peak of the first one was located in the interval of *AX-94812958 and AX-110910133* under multiple environmental conditions, and its physical position was located on the genomic interval of 8.97–21.51 Mb according to the sequence assembly of *Ae. tauschii* AL8/78 [27]. Under the five environmental conditions, this QTL explained 6.91–9.17% of the phenotypic variation (PVE). And the QTL allele from Syn80 increased the PH of seedlings, with its additive effect ranging from 1.93 to 2.92 cm (Table 6). The second QTL was located in the interval of *AX-109917900—AX-110605376* with its physical interval corresponding to 324.36–557.58 Mb in *Ae. tauschii* AL8/78. *QPh.saas-7D* explained an average PVE of 16.12% across the

**Table 5. Segregation distortion of SNP loci in RILs.**

| Chromosome | Syn80-biased Locus | | Unbiased Locus | | Syn79-biased Locus | |
|---|---|---|---|---|---|---|
| | Number | Rate (%) | Number | Rate (%) | Number | Rate (%) |
| 1D | <u>11</u> | 36.67 | 19 | 63.33 | 0 | 0.00 |
| 2D | <u>14</u> | 70.00 | 5 | 25.00 | 1 | 5.00 |
| 3D | 1 | 12.50 | 7 | 87.50 | 0 | 0.00 |
| 4D | 2 | 9.52 | 19 | 90.48 | 0 | 0.00 |
| 5D | 3 | 12.00 | 20 | 80.00 | 2 | 8.00 |
| 6D | 1 | 6.67 | 14 | 93.33 | 0 | 0.00 |
| 7D | <u>18</u> | 52.94 | 15 | 44.12 | 1 | 2.94 |
| Total | 50 | 32.68 | 99 | 64.71 | 4 | 2.61 |

Underline means genetic regions with linked loci; Chi-squared tests were considered at the P = 0.05 level

**Table 6. QTLs for plant weight (PH), tiller number (TN), shoot fresh weight (SFW) and shoot dry weight (SDW) in the RILs.**

| Traits | QTL | Environments | Peak position (cM) | Marker interval | Physical interval (Mb) | LOD | PVE (%) | ADD[¶] |
|---|---|---|---|---|---|---|---|---|
| PH | QPh.saas-1DS | 2017GHS | 1DS:34 | AX-94812958 [a] - AX-109908110 [b] | 8.97–11.57 | 4.87 | 9.17 | -1.93 |
| | | 2018GHS | 1DS:34 | AX-94812958 - AX-109908110 | 8.97–11.57 | 4.49 | 8.15 | -2.58 |
| | | 2019GHS | 1DS:34 | AX-94812958 - AX-109908110 | 8.97–11.57 | 4.86 | 6.91 | -2.31 |
| | | 2017CSS | 1DS:40 | AX-94812958 - AX-110910133 [c] | 8.97–21.51 | 2.90 | 7.71 | -2.92 |
| | | 2018CSS | 1DS:39 | AX-94812958 - AX-110910133 | 8.97–21.51 | 2.90 | 7.68 | -2.60 |
| | QPh.saas-7D | 2017GHS | 7D:90 | AX-109917900 [d] - AX-110605376 [e] | 324.36–557.58 | 7.81 | 14.64 | -2.51 |
| | | 2018GHS | 7D:91 | AX-109937582 [f] - AX-110605376 | 549.19–557.58 | 6.87 | 12.86 | -3.33 |
| | | 2019GHS | 7D:91 | AX-109937582 - AX-110605376 | 549.19–557.58 | 20.16 | 34.33 | -5.35 |
| | | 2017CSS | 7D:90 | AX-109917900 - AX-110605376 | 324.36–557.58 | 4.08 | 9.00 | -2.75 |
| | | 2018CSS | 7D:91 | AX-109937582 - AX-110605376 | 549.19–557.58 | 5.98 | 9.77 | -3.31 |
| TN | QTn.saas-1DS | 2017GHS | 1DS:33 | AX-94812958 - AX-109908110 | 8.97–11.57 | 3.36 | 6.32 | -0.73 |
| | | 2018GHS | 1DS:37 | AX-94812958 - AX-110910133 | 8.97–21.51 | 8.82 | 16.34 | -1.82 |
| | | 2019GHS | 1DS:34 | AX-94812958 - AX-109908110 | 8.97–11.57 | 12.39 | 15.54 | -1.54 |
| | | 2017CSS | 1DS:38 | AX-94812958 - AX-110910133 | 8.97–21.51 | 10.26 | 19.55 | -1.32 |
| | | 2018CSS | 1DS:35 | AX-94812958 - AX-110910133 | 8.97–21.51 | 12.93 | 18.93 | -1.42 |
| | QTn.saas-7D | 2017GHS | 7D:91 | AX-109937582 - AX-110605376 | 549.19–557.58 | 8.38 | 16.68 | -1.24 |
| | | 2018GHS | 7D:91 | AX-109937582 - AX-110605376 | 549.19–557.58 | 12.29 | 18.88 | -2.12 |
| | | 2019GHS | 7D:91 | AX-109937582 - AX-110605376 | 549.19–557.58 | 26.15 | 38.25 | -2.52 |
| | | 2017CSS | 7D:91 | AX-109937582 - AX-110605376 | 549.19–557.58 | 13.51 | 19.27 | -1.46 |
| | | 2018CSS | 7D:91 | AX-109937582 - AX-110605376 | 549.19–557.58 | 19.48 | 28.66 | -1.86 |
| SFW | QSfw.saas-1DS | 2017GHS | 1DS:33 | AX-94812958 - AX-109908110 | 8.97–11.57 | 3.70 | 7.51 | -8.98 |
| | | 2018GHS | 1DS:34 | AX-94812958 - AX-109908110 | 8.97–11.57 | 6.40 | 11.31 | -11.17 |
| | | 2019GHS | 1DS:36 | AX-94812958 - AX-110910133 | 8.97–21.51 | 7.23 | 12.63 | -10.11 |
| | | 2017CSS | 1DS:42 | AX-94812958 - AX-110910133 | 8.97–21.51 | 2.96 | 8.03 | -8.26 |
| | | 2018CSS | 1DS:36 | AX-94812958 - AX-110910133 | 8.97–21.51 | 4.27 | 8.71 | -10.30 |
| | QSfw.saas-7D | 2017GHS | 7D:91 | AX-109937582 - AX-110605376 | 549.19–557.58 | 4.83 | 9.13 | -10.13 |
| | | 2018GHS | 7D:91 | AX-109937582 - AX-110605376 | 549.19–557.58 | 9.07 | 16.25 | -13.70 |
| | | 2019GHS | 7D: 91 | AX-109937582 - AX-110605376 | 549.19–557.58 | 15.97 | 26.61 | -15.88 |
| | | 2017CSS | 7D: 91 | AX-109937582 - AX-110605376 | 549.19–557.58 | 6.03 | 10.17 | -10.93 |
| | | 2018CSS | 7D:91 | AX-109937582 - AX-110605376 | 549.19–557.58 | 7.66 | 14.08 | -14.22 |
| SDW | QSdw.saas-1DS | 2017GHS | 1DS:40 | AX-94812958 - AX-110910133 | 8.97–21.51 | 4.35 | 10.59 | -1.39 |
| | | 2018GHS | 1DS:34 | AX-94812958 - AX-109908110 | 8.97–11.57 | 6.93 | 12.28 | -1.53 |
| | | 2019GHS | 1DS:36 | AX-94812958 - AX-110910133 | 8.97–21.51 | 6.98 | 12.84 | -1.40 |
| | | 2017CSS | 1DS:40 | AX-94812958 - AX-110910133 | 8.97–21.51 | 2.73 | 7.24 | -1.02 |
| | | 2018CSS | 1DS:35 | AX-94812958 - AX-110910133 | 8.97–21.51 | 3.93 | 7.79 | -1.28 |
| | QSdw.saas-7D | 2017GHS | 7D:89 | AX-109917900 - AX-110605376 | 324.36–557.58 | 4.40 | 6.53 | -1.22 |
| | | 2018GHS | 7D:91 | AX-109937582 - AX-110605376 | 549.19–557.58 | 8.24 | 14.81 | -1.71 |
| | | 2019GHS | 7D:91 | AX-109937582 - AX-110605376 | 549.19–557.58 | 13.10 | 22.20 | -2.00 |
| | | 2017CSS | 7D:91 | AX-109937582 - AX-110605376 | 549.19–557.58 | 5.77 | 10.09 | -1.38 |
| | | 2018CSS | 7D:91 | AX-109937582 - AX-110605376 | 549.19–557.58 | 6.88 | 13.24 | -1.76 |

[a, b, c, d, e, f] indicate the Chi-square value = 38.506 (P<0.001), 57.346 (P<0.001), 6.821 (P<0.01), 76.722 (P<0.001), 79.258 (P<0.001) and 82.713 (P<0.001) for segregation distortion at these markers, respectively.

[¶]Additive effect. Positive, negative mean Syn79, Syn80 alleles produced larger values, respectively. PH: plant height, TN: tiller number, SFW; shoot fresh weight, SDW: shoot dry weight

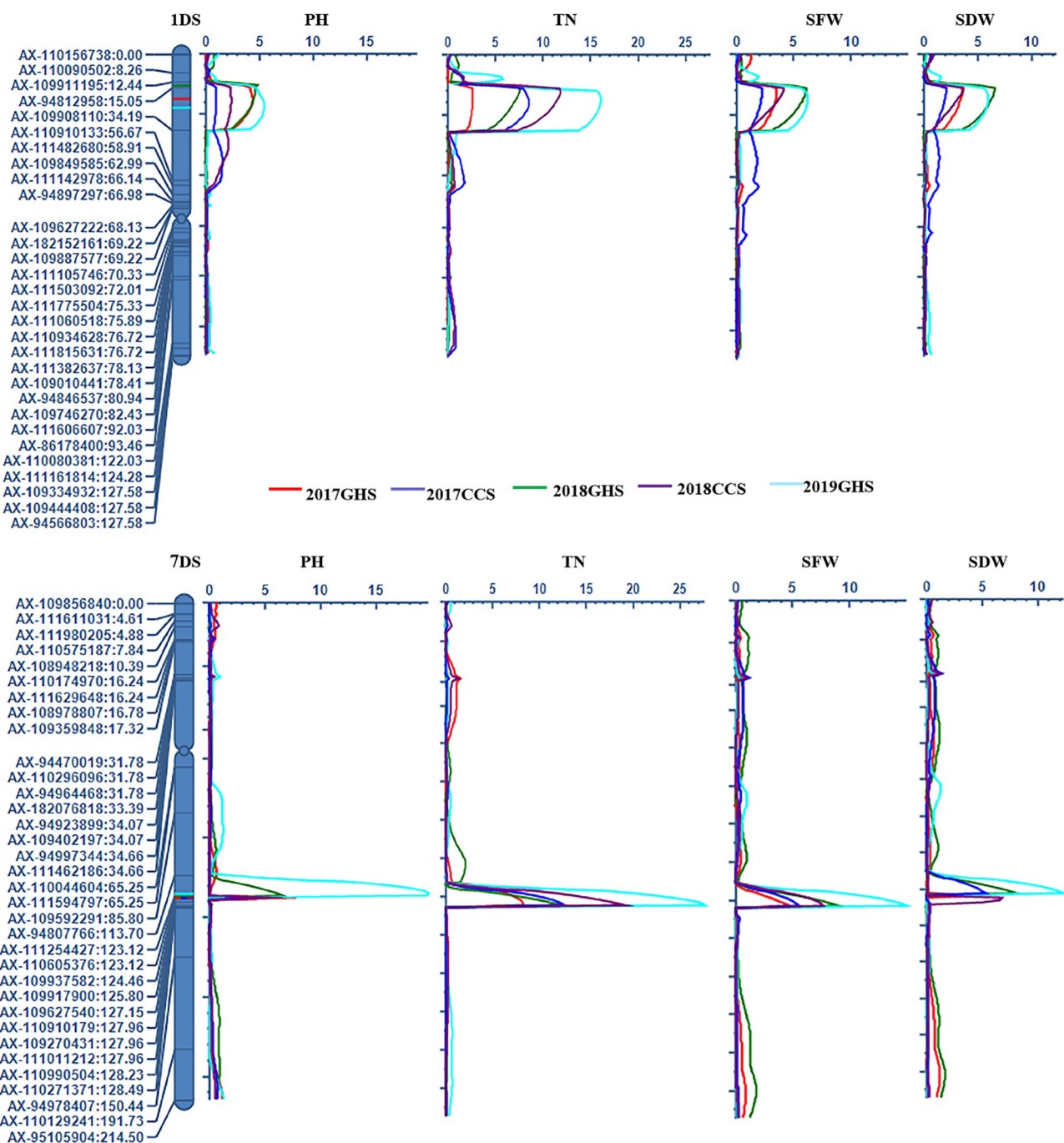

**Fig 3. QTLs for plant height (PH), tiller number (TN), shoot fresh weight (SFW) and shoot dry weight (SDW) detected on 1D and 7D in five separate trials.**

different environments. Seedling height on the QTL allele from the parent Syn80 increased more than 5 cm in the trial of 2019GHS (Table 6). For TN, two QTLs, *QTn.saas-1DS* and *QTn. saas-7D* were detected under all five environmental conditions (Table 6; Fig 3). Their intervals were in accordance with the PH QTLs on chromosome 1DS and 7D, respectively (Table 6; Fig 3). The PVE of *QTn.saas-1DS* ranged from 6.32% to 19.55% with an average of 15.34%,

and was able to increase the tiller number by about 2 tillers from Syn80 in the trial of 2018GHS (Table 6).

SDW is positively related to SFW at the seedling stage. In this study, we detected two QTLs for both SFW and SDW on the chromosome 1DS and 7D (Table 6; Fig 3). The QTL intervals for SFW and SDW were in accordance with the QTL intervals for both PH and TN. Since higher plant height and a greater tiller number per plant resulted in larger SFW and SDW, this suggests that these may be the same QTLs. The average PVE for *QSfw.saas-1DS* and *QSdw. saas-1DS* was 9.64% and 10.15%, respectively. The QTL allele from Syn80 increased the SFW and SDW (Table 6). In the interval of *AX-109937582—AX-110605376* on chromosome 7D, QTLs for both SFW and SDW were identified under all five environmental conditions, and the average PVE of *QSfw.saas-7D* and *QSdw.saas-7D* was 15.25% and 13.37%, respectively (Table 6). The QTL alleles that increased SFW and SDW were from the parent Syn80 (Table 6).

In this study, two genomic regions were identified to be associated with early biomass. They were in the interval of *AX-94812958—AX-110910133* on chromosome 1DS and the interval of *AX-109917900—AX-110605376* on chromosome 7D. The two genomic regions from the parent Syn80 could significantly enhance the early biomass with pleiotropic effects of increasing PH, TN, SFW and SDW.

## Discussion

Greater early biomass is visual and important for breeding new varieties and innovative utilization of crop germplasm, especially under adverse environmental conditions. Therefore, it is important to select traits under drought stress [29–31], especially in Sichuan, where drought or seasonal drought occurred frequently in the last 70 years [32]. In this study, the early biomass of the parents and the RIL population showed significant phenotypic differences in PH, TN, SFW and SDW under the five different environmental conditions from 2016 to 2019. Phenotypic and QTL analyses demonstrated that the early biomass related traits, PH, TN, SFW and SDW, were controlled by polygenes. Wheat growth habit types (spring or winter), the wheat growth progress and early biomass were affected by the combination of photoperiod and vernalization genes [33–36]. Photoperiod and vernalization genes on the D genome were located on 2D and 5D [33,34]. Considering that the phenology and growing periods of the two parents and the RILs were slight different, it can be inferred that early biomass in these RILs was controlled by genes, which could not be related to photoperiod or vernalization genes, for no QTLs were detected on the chromosomes 2D or 5D.

In the present study, two synthetic wheat varieties, Syn79 and Syn80, were generated from two different *Ae. tauschii* accessions crossed with the same tetraploid wheat, and the significant difference in early biomass between them was caused by their different D genome donors. *Ae. tauschii*, the D genome donor of common wheat, exhibited genetic diversity for early growth and might be a valuable species for improvement of early vigour in wheat [37]. The common wheat D genome progenitor, *Ae. tauschii*, showed a rapid leaf expansion rate at the seedling stage [21,37], which is beneficial for reducing evaporative losses from the soil [21]. Genetic dissection for early vigour related traits has been reported in several germplasms under different growing conditions, and QTLs for early vigour related traits were distributed through almost the whole genome of the wheat [21,37–41]. ter Steege et al identified 87 QTLs for early growth that were related to 33 traits, 3.1 QTLs per trait, explaining 32% of the PVE by using a population of *Ae. tauschii* RILs at the seedling stage, but there was no significant QTLs for plant and shoot mass detected in this study, considering that the effects of QTL for the underlying growth traits counterbalanced each other [37]. However, in our study, two chromosome

fragments for SFW and SDW were detected, which simultaneously regulated PH and TN. The favorable alleles detected were from *Ae. tauschii* and they could express stably in a hexaploid genetic background. Few QTLs for biomass have been identified in the diploid populations of *Ae. tauschii* [37], but in a hexaploidy genetic background. In the present study these expressed stably in synthetic hexaploid wheat. The AABB genome of tetraploid wheat may play a very important role in synthetic wheat derived from crosses of tetraploid wheat and *Ae. tauschii*. The effects of genome combination between AABB and DD for gene expression need to be analyzed further. And it substantiates the conclusion that using SHW is a more effective method to transfer favourable genes from *Ae. tauschii* to common wheat [6,7,9,42].

In addition, the two chromosome fragments for PH, TN, SFW and SDW were detected stably on 1DS and 7D, which were located on the genomic intervals of 8.98–21.51 Mb and 324.36–557.58 Mb, respectively. *Lr42*, *Rmg6*, *Sr33*, *SrTA1662*, *LR10*, *Xa5*, *Chalk5*, *MHZ5*, *B10*, *Rc*, *BC10*, *EBR1* and *EBR1* were located in the interval of *AX-94812958* -*AX-110910133* on 1DS of *Ae. tauschii*, and 16 QTL/genes (*Pid2*, *IPA1*, *Xa13*, *Hd18*, *GW8*, *Xa27-Xa27-IRBB27*, *qUVR-10*, *Yr33*, *Dn2*, *Ehd3*, *Nud*, *OsABCG15*, *MOC1*, *Lks2*, *TaD27* and QTls for antixenosis) were in the interval of *AX-109917900* -*AX-110605376* on 7D [43]. Among these reported genes, none except for *TaD27* on 7D, which was associated with tiller number in hexaploidy, has been found to be related to early vigour previously.

Segregation distortion is a common phenomenon among many plants [44]. In the present study, 54 of 153 SNP loci showed segregation distortion in the RILs, and 50 makers were skewed to Syn80, while 4 were biased to Syn79. Segregation distortion loci accounted for 35.29% of the total polymorphic loci, and 92.59% of the loci were preferentially biased to the female parent Syn80, with only 7.41% coming from male parent Syn79. At the same time, we found that Syn80 had stronger seedling vigour than that of Syn79. Therefore, the early vigour which afforded a high survival ratio in the RILs containing the Syn80 loci, was higher than that of the RILs containing the Syn79 loci. The proportion of segregation distortion was high in the RILs. Xu et al found a similar phenomenon, finding that the purer the population, the higher separation ratio [45]. In the present study, three genomic regions were detected to be Syn80--biased on chromosome 1D, 2D and 7D (Table 5), which were involved with the QTL intervals for early biomass. The centre of segregation bias on chromosome 1DS was located in the interval of *AX-110090502—AX-109911195* with a genetic location from 8.26 cM to 12.44 cM, as 96.4% of the progeny shared the same genotype with the parent Syn80 at the SNP site of *AX-110090502* and 97.9% for *AX-109911195*. On chromosome 2D, the segregation bias region was framed from *AX-108911375* to *AX-110935958* across about 20 cM. On 7D, the centre of segregation bias was located in the interval of *AX-110271371* to *AX-94807766*. The centre of segregation bias on 1DS was about 25 cM away from the detected QTL peaks for early growth-related traits, and the centre of segregation bias on 7D was located in the interval of the QTL peaks detected on this chromosome. Many factors may cause segregation distortion, these can be genetic factors such as reproductive isolation, or incoordination between the cytoplasm and nucleus, or hybrid necrosis etc. [46], and these can be due to natural or artificial selection [47]. In most cases, segregation is controlled by reproductive isolation factors such as gametophyte genes on the nucleus or sterility genes [48–51]. Several types of hybrid abnormalities including hybrid necrosis were reported in the process of synthetic wheat production [52,53]. Usually, these abnormal growth phenotypes are classified into hybrid necrosis (Types II and III), hybrid chlorosis and severe growth abortion [54,55]. Two genes derived from *Ae.tauschii* related to type II and III necrosis symptoms have been mapped [53,54]. The gene *Nec1* of type III necrosis was on chromosome 7DS [54], while the gene *Nec2* of type II necrosis was on chromosome 2DL [56]. The locations of *Nec2* and *Nec1* were close to the segregation bias region on chromosome 2D and the segregation bias centre on chromosome 7D. One possible reason for the

segregation bias for Syn80 in these loci was that Syn79 may have carried the *Nec2* and *Nec1* alleles for hybrid necrosis. Thus, the segregation bias would have spread from the location of *Nec2* or *Nec1* across the QTL regions in this population. Segregation distortion regions may be related to certain genes, the gene location of the target trait can be preliminarily determined according to the segregation distortion region of the genetic map and the phenotypic data. However, no strong evidence showed that the early biomass QTL was caused by the segregation bias to Syn80.

In the present study, 3480 SNP markers were used on the D genome, and only 153 polymorphic markers were detected between the parents, a percentage polymorphism of 4.40%. Comparing to the genetic diversity of *Ae. taushcii* and the wheat cultivars reported by previous authors [56–58], Syn79 and Syn80 had low genetic diversity on the D genome. It has been widely accepted that *Ae. tauschii* ssp. *strangulata* is the D genome donor of hexaploid wheat [56,59–63]. *Ae. tauschii* was classified into two groups, lineage 1 and lineage 2 [56,64]. Lineage 1 is broadly related to *Ae. tauschii* ssp. *tauschii* and lineage 2 is broadly related to *Ae. tauschii* ssp. *strangulata*. The Infinium SNP array for the D genome was developed mainly according to the SNP polymorphism between *Ae. tauschii* ssp. *tauschii* and *Ae. tauschii* ssp. *strangulata*. Therefore, the D genome donors (AT333 and AT428) of synthetic hexaploid wheat Syn79 and Syn80 may belong to the same group (Lineage 1 or Lineage 2), and their genetic relationship is very close. Although the number of polymorphic loci in the D genome between Syn79 and Syn80 was low, two genome regions on 1DS and 7D for four early biomass related traits were still detected under five different environmental conditions. This provided a basis for further fine mapping and candidate gene analysis of a few QTLs for early biomass related traits. On the other hand, each of the synthetic wheat Syn79 and Syn80 combining elite genes from tetraploid wheat and *Ae. tauschii* is a potential resource to broaden the genetic diversity for wheat breeding programs.

## Conclusion

By using a set of recombinant inbred lines derived from two synthetic hexaploid wheat varieties (Syn79 and Syn80) re-synthesized from the same tetraploid wheat DOY1 and two different *Ae. tauschii* accessions (AT333 and AT428), two genomic regions on 1DS and 7D were detected to be associated with early biomass, with pleiotropic effects on PH, TN, SFW and SDW. The QTL alleles from Syn80 enhanced the early biomass by increasing PH, TN, SFW and SDW, and these originated from the *Ae. tauschii* AT428, which expresses stably in a hexaploid background. The framed SNP markers could be used for wheat improvement.

## Supporting information

**S1 Data.**
(XLS)

**S2 Data.**
(XLS)

**S3 Data.**
(XLS)

## Acknowledgments

We thank the International Maize and Wheat Improvement Center for providing synthetic hexaploid wheat parents (Syn79 and Syn80). The National Natural Science Foundation of

China and the Department of Science and Technology of Sichuan Province supported this work. We also thank our colleagues and staff at the Guang-Han Station and the Cang-shan Station of the Sichuan Academy of Agricultural Sciences for their helps in this study.

## Author Contributions

**Conceptualization:** Long Mao, Yonghong Zhou.

**Data curation:** Yumin Yang, Hongshen Wan, Yonghong Zhou.

**Formal analysis:** Yumin Yang, Hongshen Wan.

**Funding acquisition:** Aili Li, Long Mao, Wuyun Yang.

**Investigation:** Yumin Yang, Fan Yang, Chun Xiao, Meijin Ye, Chunxiu Chen, Guangmin Deng, Qin Wang.

**Methodology:** Yumin Yang, Wuyun Yang, Yonghong Zhou.

**Project administration:** Jun Li.

**Resources:** Jun Li.

**Software:** Hongshen Wan, Fan Yang, Qin Wang.

**Writing – original draft:** Yumin Yang.

**Writing – review & editing:** Hongshen Wan, Wuyun Yang, Yonghong Zhou.

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
