## [Decision Letter · Decision Letter 0]

13 Mar 2020

PONE-D-20-04315

Mapping QTL for enhancing early biomass derived from Aegilops tauschii in synthetic hexaploid wheat

PLOS ONE

Dear Dr Yang,

Thank you for submitting your manuscript to PLOS ONE. After careful consideration, we feel that it has merit but does not fully meet PLOS ONE’s publication criteria as it currently stands. Therefore, we invite you to submit a revised version of the manuscript that addresses the points raised during the review process.

We would appreciate receiving your revised manuscript by Apr 27 2020 11:59PM. To enhance the reproducibility of your results, we recommend that if applicable you deposit your laboratory protocols in protocols.io, where a protocol can be assigned its own identifier (DOI) such that it can be cited independently in the future. For instructions see: http://journals.plos.org/plosone/s/submission-guidelines#loc-laboratory-protocols

We look forward to receiving your revised manuscript.

Kind regards,

Aimin Zhang, Ph.D.

Academic Editor

PLOS ONE

Journal Requirements:

2) We suggest you thoroughly copyedit your manuscript for language usage, spelling, and grammar. If you do not know anyone who can help you do this, you may wish to consider employing a professional scientific editing service.  

3) PLOS requires an ORCID iD for the corresponding author in Editorial Manager on papers submitted after December 6th, 2016. Please ensure that you have an ORCID iD and that it is validated in Editorial Manager. To do this, go to ‘Update my Information’ (in the upper left-hand corner of the main menu), and click on the Fetch/Validate link next to the ORCID field. This will take you to the ORCID site and allow you to create a new iD or authenticate a pre-existing iD in Editorial Manager. Please see the following video for instructions on linking an ORCID iD to your Editorial Manager account: https://www.youtube.com/watch?v=_xcclfuvtxQ

Reviewers' comments:

Reviewer's Responses to Questions

**Comments to the Author**

1. Is the manuscript technically sound, and do the data support the conclusions?

Reviewer #1: Yes

Reviewer #2: Yes

2. Has the statistical analysis been performed appropriately and rigorously? 

Reviewer #1: Yes

Reviewer #2: Yes

3. Have the authors made all data underlying the findings in their manuscript fully available?

Reviewer #1: Yes

Reviewer #2: Yes

4. Is the manuscript presented in an intelligible fashion and written in standard English?

Reviewer #1: Yes

Reviewer #2: No

5. Review Comments to the Author

Reviewer #1: This manuscript described analysis of QTLs related to early biomass using a RIL population derived from two synthetic hexaploid wheat with same AABB-genome background. Four stable QTL for plant height (PH), tiller number (TN), shoot fresh weight (SFW) and shoot dry weight (SDW) were detected on chromosomes 1D and 7D. The manuscript presented some interesting results using synthetic hexaploid wheat accessions distinguished by only D genome. I felt this manuscript is valuable because that diversity of wheat D genome is very low. The introduction of novel D-derived genes through synthetic hexaploid is a efficient way to expand genetic background of bread wheat. The identification of novel QTLs for early biomass from Aegilops tauschii are valuable for wheat improvement. It will be of interest for the readers of Plos one. Thus, this manuscript is acceptable, but there are some problem needs to be addressed.

1)There are some witting errors needs thorough correction, some were marked on the pdf. And the language also need improve.

2)Some statistical information need to reconfirm and replenish.

P10-L181-183: The authors claim that the H2 =0.4327 for PH is significant lower than the results of previous reported. I suggested that the authors perform an ANOVA again to confirm whether the error variance is too large or the calculation is wrong.

P13-L221-228.: The genetic map constructed in present study containing 54 segregation distortion loci. I suggested that the authors confirm whether these segregation distortion loci have an effect on QTL mapping.

P19-L290-293: The authors claim that the growing periods of two parents and RILs were consistent, but I found this claim was not supported by the presented data.

Reviewer #2: This manuscript reports important chromosomal positions of QTLs for four biomass traits that affect wheat yield and tolerance to abiotic stresses. It would be worthy of being published if the numerous typographical and grammatical errors (identified in the uploaded file) are corrected throughout the manuscript. It is also suggested that authors modify the figure of chromosomes 1D and 7D to depict the centromeres.

6. PLOS authors have the option to publish the peer review history of their article (what does this mean?). If published, this will include your full peer review and any attached files.

Reviewer #1: No

Reviewer #2: Yes: Richard R.-C. Wang

---

## [Author Response · Author response to Decision Letter 0]

31 Mar 2020

Thank you very much for giving us an opportunity to revise our manuscript, we appreciate editor and reviewers very much for their positive and constructive comments and suggestions on our manuscript. We have accepted all the modifications made by the two reviewers, and answered the comments carefully. We have tried our best to revise our manuscript according to the reviewers’ comments and PLOS ONE’s style requirements. In addition, this manuscript was edited for proper English language, grammar, punctuation, spelling, and overall style by one or more of the highly qualified native English speaking editors at NativeEE. We have made revision which marked in red in the paper.

Point-to-point reply:

Response: 

Thanks for your reminder. We carefully have read PLOS ONE’s style requirements again, and carefully modified to meet its style requirements.

2) We suggest you thoroughly copyedit your manuscript for language usage, spelling, and grammar.

Response: 

Thanks for your suggestion. Native English speaking editors at Native English Editing checked and corrected for English language, spelling, grammar.

The name of the colleague or the details of the professional service that edited your manuscript

A copy of your manuscript showing your changes by either highlighting them or using track changes (uploaded as a *supporting information* file)

A clean copy of the edited manuscript (uploaded as the new *manuscript* file)

Response: 

OK. We will resubmit the manuscript according to your requirements.

Our manuscript for language usage, spelling, and grammar were checked and corrected by native English speaking editors at Native English Editing, which provided a statement of editing. We uploaded as separate file and labeled ‘Statement of Editing’.

We showed our changes in our manuscript by in red font. The file was uploaded separately and labeled ‘Revised Manuscript with Track Changes’.

We uploaded a clean manuscript separately, this file was labeled ‘Manuscript’.

3) PLOS requires an ORCID iD for the corresponding author in Editorial Manager on papers submitted after December 6th, 2016. Please ensure that you have an ORCID iD and that it is validated in Editorial Manager. To do this, go to ‘Update my Information’ (in the upper left-hand corner of the main menu), and click on the Fetch/Validate link next to the ORCID field. This will take you to the ORCID site and allow you to create a new iD or authenticate a pre-existing iD in Editorial Manager. Please see the following video for instructions on linking an ORCID iD to your Editorial Manager account: https://www.youtube.com/watch?v=_xcclfuvtxQ

Response:

OK. The corresponding author Wuyun Yang has an ORCID iD, and it is validated in Editorial Manager.

Reviewers' comments:

Reviewer's Responses to Questions

Comments to the Author

1. Is the manuscript technically sound, and do the data support the conclusions?

Reviewer #1: Yes

Reviewer #2: Yes

Response:

Thanks for your recognition.

2. Has the statistical analysis been performed appropriately and rigorously?

Reviewer #1: Yes

Reviewer #2: Yes

Response:

Thanks for your recognition.

3.Have the authors made all data underlying the findings in their manuscript fully available?

Reviewer #1: Yes

Reviewer #2: Yes

Response: 

Thanks for your recognition.

4. Is the manuscript presented in an intelligible fashion and written in standard English?

Reviewer #1: Yes

Reviewer #2: No

Response:

We have tried our best to modify the manuscript. At the same time, this manuscript was edited for proper English language, grammar, punctuation, spelling, and overall style by one or more of the highly qualified native English speaking editors at NativeEE. We hope that the revised manuscript can meet PLOS ONE’s requirements.

5. Review Comments to the Author

Reviewer #1: This manuscript described analysis of QTLs related to early biomass using a RIL population derived from two synthetic hexaploid wheat with same AABB-genome background. Four stable QTL for plant height (PH), tiller number (TN), shoot fresh weight (SFW) and shoot dry weight (SDW) were detected on chromosomes 1D and 7D. The manuscript presented some interesting results using synthetic hexaploid wheat accessions distinguished by only D genome. I felt this manuscript is valuable because that diversity of wheat D genome is very low. The introduction of novel D-derived genes through synthetic hexaploid is a efficient way to expand genetic background of bread wheat. The identification of novel QTLs for early biomass from Aegilops tauschii are valuable for wheat improvement. It will be of interest for the readers of Plos One. Thus, this manuscript is acceptable, but there are some problem needs to be addressed.

1)There are some witting errors needs thorough correction, some were marked on the pdf. And the language also need improve.

Response:

Thank you for your careful revision. We have accepted your correction. We have tried our best to modify the manuscript. At the same time, this manuscript was edited for proper English language, grammar, punctuation, spelling, and overall style by one or more of the highly qualified native English speaking editors at NativeEE. We hope that the revised manuscript can meet PLOS ONE’s requirements.

2)Some statistical information need to reconfirm and replenish.

P10-L181-183: The authors claim that the H2 =0.4327 for PH is significant lower than the results of previous reported. I suggested that the authors perform an ANOVA again to confirm whether the error variance is too large or the calculation is wrong.

Response:

In this study, the PH was investigated at the jointing stage. The method of measuring PH was different from the measure of PH in the mature period, which is the length of the main stem, measured from ground level to the tip of spike, excluding awns. However, the PH of seedling in this study was measured from ground level to the tip of the first leaf. We infer that the lower H2 for PH in this study was caused by the alterable length of the first leaf, as H2 for the length of wheat leaf was much lower that the PH in the mature period. Actually, it is hard to investigated the plant height before heading time.

Plant height is a complex trait, and is controlled by multiple major genes and micromajor genes. The QTLs detected in our research were micromajor genes, their heritability was lower, compared to the major genes.

We performed an ANOVA again, as was shown in table 1 below. The ANOVA results showed that variation of plant height was determined by genotype, environment and genotype × environment interactions (Table 2 in manuscript). Anyway, the lower H2 for PH in this study could be considered to be caused by the multi-environments.

Table 1 - ANOVA analysis of RIL population in five different field environments.

 Trait Sum of squares df Mean square F Sig.

Environment PH(cm) 156917.231 4 39229.308 1368.743 .000

 TN(No./plant) 15260.574 4 3815.143 780.237 .000

 SFW(g/plant) 249927.824 4 62481.956 144.401 .000

 SDW(g/plant) 4629.493 4 1157.373 161.466 .000

Genotype PH(cm) 81328.358 202 402.616 14.048 .000

 TN(No./plant) 14145.042 202 70.025 14.321 .000

 SFW(g/plant) 1227777.367 202 6078.106 14.047 .000

 SDW(g/plant) 121740.517 202 602.676 84.080 .000

Environment×genotype PH(cm) 46743.096 808 57.850 2.018 .000

 TN(No./plant) 8113.498 808 10.041 2.054 .000

 SFW(g/plant) 756089.686 808 935.755 2.163 .000

 SDW(g/plant) 38235.366 808 47.321 6.602 .000

Error PH(cm) 52363.345 1827 28.661 

 TN(No./plant) 8933.524 1827 4.890 

 SFW(g/plant) 790535.902 1827 432.696 

 SDW(g/plant) 13095.739 1827 7.168 

Total PH(cm) 8540199.707 2842 

 TN(No./plant) 460303.452 2842 

 SFW(g/plant) 13613684.720 2842 

 SDW(g/plant) 357908.221 2842 

P13-L221-228.: The genetic map constructed in present study containing 54 segregation distortion loci. I suggested that the authors confirm whether these segregation distortion loci have an effect on QTL mapping.

Response:

In this study, 54 segregation distortion loci were detected, but these segregation distortion loci don’t affect QTL mapping. If the high density map was constructed accurately, and then the impact of segregation distortion on QTL analysis can be ignored. QTL IciMapping software can be used to construct genetic map, QTL mapping in segregation distortion population. And we checked the segregation ratio of the loci framed the QTLs and no significant segregation distortion was found. And the relevant results were described in the discussion of the manuscript (P21-L343-346 ans P22-L361-364).

P19-L290-293: The authors claim that the growing periods of two parents and RILs were consistent, but I found this claim was not supported by the presented data.

Response:

Sorry, the expression of this sentence is not accurate enough. Actually, the difference of growing periods between two parents and RILs exist, but was very small. And we revised the sentence, changed ‘consistent’ into ‘slight different’.

Reviewer #2: This manuscript reports important chromosomal positions of QTLs for four biomass traits that affect wheat yield and tolerance to abiotic stresses. It would be worthy of being published if the numerous typographical and grammatical errors (identified in the uploaded file) are corrected throughout the manuscript. It is also suggested that authors modify the figure of chromosomes 1D and 7D to depict the centromeres.

Response:

Thank you for your careful revision, suggestion on our manuscript, and recognition of our work. We have accepted your correction. We have tried our best to modify the manuscript. At the same time, this manuscript was edited for proper English language, grammar, punctuation, spelling, and overall style by one or more of the highly qualified native English speaking editors at NativeEE.

We have modified the figure of 1D and 7D, and attached the position of the centromeres (Fig 3).

6. PLOS authors have the option to publish the peer review history of their article (what does this mean?). If published, this will include your full peer review and any attached files.

Do you want your identity to be public for this peer review? For information about this choice, including consent withdrawal, please see our Privacy Policy.

Reviewer #1: No

Reviewer #2: Yes: Richard R.-C. Wang

Response:

Yumin Yang register a user(362072749@qq.com), and upload our the three figure files, they are valid TIF files.

Response to comments in manuscript by reviewer #1：

1.P9-L169-L171：”Independent of the differences between the two parents, in all trials there were significant variations in the investigated traits of the RIL populations, with values spanning much larger ranges than those defined by the parental values.” This sentence is little confusing.

Response:

We want to state the transgressive inheritance. so we added it to avoid confusion (P9-L171-L174).

2. P13-L212:”and then aligned by nnTwoOpt method.” ref should be added.

Response:

This sentence’s ref were added (P13-L214).

3.P20-L310：”Few QTLs for biomass have been identified in the diploid populations directly from Ae. tauschii, but in a hexaploidy genetic background.”

results or refs supporting this should be added.

Response:

We want to state that ter Steege et al haven’t detected QTLs for biomass in diploid populations of Ae. tauschii, but we detected in synthetic hexaploid wheat. This sentence added ref, and the sentence changed into “Few QTLs for biomass have been identified in the diploid populations of Ae. tauschii [37], but in a hexaploidy genetic background.”(P20-L313-L315).

Response to comments in manuscript by reviewer #2：

1. P7-L133-L135:The RILs and their parents Syn79 and Syn80 were executed on the Affymetrix platform of Axiom Wheat Breeder’s Genotyping Array with 13947 SNP markers including 1272 functional markers by China Golden Marker Biotech Co Ltd (Beijing, China). 

“executed” need a better word.

Response:

We changed “executed” into “genotyped” (P7-L134)

2. P23-L380-L382: On the other hand, each of the synthetic wheat Syn79 and Syn80 having newly genes from tetraploid wheat and Ae. tauschii is a potential resource to broaden the genetic diversity for wheat breeding programs.

Newly acquired? Any evidence for the newly acquired genes in SHWs resulted from gene recombination following hybridization of tetraploid wheat and Ae. tauschii?

Response:

Sorry, the expression of “newly genes’ is not accurate enough, because we have no relevant evidence to confirm whether these genes have been reported. We can make sure that Syn80 has QTLs for enhancing early biomass in this study, with pleiotropic effects on plant height, tiller number, shoot fresh weight and shoot dry weight. These genes were elite genes from tetraploid wheat and Ae. tauschii. So we revised this sentence, and changed ‘newly genes’ into ‘ combining elite genes’ (P23-L386).

---

## [Decision Letter · Decision Letter 1]

30 Apr 2020

PONE-D-20-04315R1

Mapping QTLs for enhancing early biomass derived from Aegilops tauschii in synthetic hexaploid wheat

PLOS ONE

Dear Dr Yang,

Thank you for submitting your manuscript to PLOS ONE. After careful consideration, we feel that it has merit but does not fully meet PLOS ONE’s publication criteria as it currently stands. Therefore, we invite you to submit a revised version of the manuscript that addresses the points raised during the review process.

We would appreciate receiving your revised manuscript by Jun 14 2020 11:59PM. To enhance the reproducibility of your results, we recommend that if applicable you deposit your laboratory protocols in protocols.io, where a protocol can be assigned its own identifier (DOI) such that it can be cited independently in the future. For instructions see: http://journals.plos.org/plosone/s/submission-guidelines#loc-laboratory-protocols

We look forward to receiving your revised manuscript.

Kind regards,

Aimin Zhang, Ph.D.

Academic Editor

PLOS ONE

Reviewers' comments:

Reviewer's Responses to Questions

**Comments to the Author**

1. If the authors have adequately addressed your comments raised in a previous round of review and you feel that this manuscript is now acceptable for publication, you may indicate that here to bypass the “Comments to the Author” section, enter your conflict of interest statement in the “Confidential to Editor” section, and submit your "Accept" recommendation.

Reviewer #1: All comments have been addressed

Reviewer #3: All comments have been addressed

2. Is the manuscript technically sound, and do the data support the conclusions?

Reviewer #1: Yes

Reviewer #3: Yes

3. Has the statistical analysis been performed appropriately and rigorously? 

Reviewer #1: Yes

Reviewer #3: Yes

4. Have the authors made all data underlying the findings in their manuscript fully available?

Reviewer #1: Yes

Reviewer #3: Yes

5. Is the manuscript presented in an intelligible fashion and written in standard English?

Reviewer #1: Yes

Reviewer #3: No

6. Review Comments to the Author

Reviewer #1: As all questions have been addressed, I felt that this manuscript could be accepted for publication.

Reviewer #3: This paper described the mapping studies of QTLs related to early biomass using a RIL population derived from two synthetic hexaploid accessions with the same tetraploid background. Two major QTLs for four biomass related traits were detected on chromosomes 1DS and 7D. The data was interesting and should provide us with useful information on expanding the diversity of wheat D genome. However, a minor revision is needed for publication.

Some points are:

1. The growing periods of RILs were the controlling factors for precise evaluation of the biomass related phenotypes, because the jointing stage was a relatively broad description. Please clarify it in M & M, for example, how many days after sowing to start phenotype evaluation, and how many days needed for finishing the plant height and tiller number evaluation and sampling for all the RILs?

2. About the loci with segregation distortion. Please add the Chi-squared test of the most significant SNPs for each QTL in Table 6.

3. The language also need to be improved, for example, in line 43, change ‘the QTLs’ to ‘these two QTLs’; line 171, ‘There was transgressive inheritance.’; line 178, ‘these genes’ phenotypic traits’; line 193, ‘the gene related to early biomass has pleiotropic effects on all four traits.’; line 215-216, ‘There were no genetic gaps, with adjacent marker separation of no more than 50 cM occurring in each chromosome’; line 329, ‘indicating that QTLs for early biomass on 1DS in this population were new genes located in this study’.

4. Please modify the Figure legends and Table notes to make it easier for readers to understand. For example, line 264, ‘Positive and negative Syn79 and Syn80 alleles produced larger and smaller values, respectively.’

5. It will be good to add the phenotypic data and pictures of AT428 and AT333.

7. PLOS authors have the option to publish the peer review history of their article (what does this mean?). If published, this will include your full peer review and any attached files.

Reviewer #1: No

Reviewer #3: No

---

## [Author Response · Author response to Decision Letter 1]

23 May 2020

Point-to-point reply to Reviewer #3: 

1. The growing periods of RILs were the controlling factors for precise evaluation of the biomass related phenotypes, because the jointing stage was a relatively broad description. Please clarify it in M & M, for example, how many days after sowing to start phenotype evaluation, and how many days needed for finishing the plant height and tiller number evaluation and sampling for all the RILs?

Response: Actually, the mature period of each RIL was mostly similar (often much later than the local cultivars), and the difference between them was slight (Fig. S1-directly photographed from the field at May-11 2020). So, the phenotype evaluation could be collected at one time, as the jointing stage of them were also similar. In this study, Phenotypic data was investigated about 110 days after sowing, when the first internode came out, the plant height and tiller number evaluation were finished during sampling within 1-2 day. Thank you for your careful revision, this should be stated clearly in M & M.

Fig. S1 RILs planted in filed on 2020.

2. About the loci with segregation distortion. Please add the Chi-squared test of the most significant SNPs for each QTL in Table 6.

Response: We performed the Chi-squared test, as was shown in Table 6. For AX-94812958, the Chi-Square value is 38.506 (P<0.001). Chi-Square of AX-109908110 is 57.346 (P<0.001). the Chi-Square value of AX-110910133 is 6.821 (P<0.01). the Chi-Square value of AX-110605376 is 79.258 (P<0.001). the Chi-Square value of AX-109937582 is 82.713 (P<0.001). the Chi-Square of AX-109917900 is 76.722 (P<0.001). 

3 and 4. The language also need to be improved, for example, in line 43, change ‘the QTLs’ to ‘these two QTLs’; line 171, ‘There was transgressive inheritance.’; line 178, ‘these genes’ phenotypic traits’; line 193, ‘the gene related to early biomass has pleiotropic effects on all four traits.’; line 215-216, ‘There were no genetic gaps, with adjacent marker separation of no more than 50 cM occurring in each chromosome’; line 329, ‘indicating that QTLs for early biomass on 1DS in this population were new genes located in this study’. 4. Please modify the Figure legends and Table notes to make it easier for readers to understand. For example, line 264, ‘Positive and negative Syn79 and Syn80 alleles produced larger and smaller values, respectively.’

Response: Question 3 and 4 have been fixed in the text.

5. It will be good to add the phenotypic data and pictures of AT428 and AT333.

Response: Unfortunately, we did not introduce the AT428 and AT333 from CIMMYT. However, considering that synthetic hexaploid wheat Syn80 and Syn79 had the same tetraploid parent, their difference of early vigour originated from their diploid parent AT428 and AT333, and the early vigour of AT428 was supposed to be greater than AT333.

Note: The above-mentioned Fig S1 and Table can be found in the attached word file “Response to Reviewers”.

---

## [Decision Letter · Decision Letter 2]

4 Jun 2020

Mapping QTLs for enhancing early biomass derived from Aegilops tauschii in synthetic hexaploid wheat

PONE-D-20-04315R2

Dear Dr. Yang,

We’re pleased to inform you that your manuscript has been judged scientifically suitable for publication and will be formally accepted for publication once it meets all outstanding technical requirements.

Kind regards,

Aimin Zhang, Ph.D.

Academic Editor

PLOS ONE

Additional Editor Comments (optional):

Reviewers' comments:

Reviewer's Responses to Questions

**Comments to the Author**

1. If the authors have adequately addressed your comments raised in a previous round of review and you feel that this manuscript is now acceptable for publication, you may indicate that here to bypass the “Comments to the Author” section, enter your conflict of interest statement in the “Confidential to Editor” section, and submit your "Accept" recommendation.

Reviewer #3: All comments have been addressed

2. Is the manuscript technically sound, and do the data support the conclusions?

Reviewer #3: Yes

3. Has the statistical analysis been performed appropriately and rigorously? 

Reviewer #3: Yes

4. Have the authors made all data underlying the findings in their manuscript fully available?

Reviewer #3: Yes

5. Is the manuscript presented in an intelligible fashion and written in standard English?

Reviewer #3: Yes

6. Review Comments to the Author

Reviewer #3: As all comments have been addressed, I suggest that this manuscript can be accepted for publication.

7. PLOS authors have the option to publish the peer review history of their article (what does this mean?). If published, this will include your full peer review and any attached files.

Reviewer #3: No

---

## [Editor Report · Acceptance letter]

10 Jun 2020

PONE-D-20-04315R2 

Mapping QTLs for enhancing early biomass derived from Aegilops tauschii in synthetic hexaploid wheat 

Dear Dr. Yang:

I'm pleased to inform you that your manuscript has been deemed suitable for publication in PLOS ONE. Congratulations! Your manuscript is now with our production department. 

Kind regards, 

on behalf of

Prof. Aimin Zhang 

Academic Editor

PLOS ONE